# Understanding Physical Activity Behavior in Ghanaian Adults with Type 2 Diabetes: A Qualitative Descriptive Study

**DOI:** 10.3390/jfmk8030127

**Published:** 2023-09-05

**Authors:** Mohammed Amin, Debra Kerr, Yacoba Atiase, Yusif Yakub, Andrea Driscoll

**Affiliations:** 1Centre for Quality and Patient Safety, Institute for Health Transformation, Faculty of Health, School of Nursing and Midwifery, Deakin University, 221 Burwood Highway, Melbourne, VIC 3125, Australia; d.kerr@deakin.edu.au (D.K.); andrea.driscoll@deakin.edu.au (A.D.); 2National Diabetes Management and Research Centre, Korle-Bu Teaching Hospital, University of Ghana Medical School, Accra P.O. Box GP 4236, Ghana; yatiase@ug.edu.gh; 3Faculty of Medicine and Health, The University of Sydney, Science Road, Sydney, NSW 2050, Australia; yyak3902@uni.sydney.edu.au

**Keywords:** physical activity, exercise, type 2 diabetes, barriers, facilitators, self-care, Ghana, adults, qualitative study

## Abstract

Despite a relatively low prevalence rate, sub-Saharan Africa bears a substantial diabetes burden. Physical activity (PA) plays a crucial role in managing type 2 diabetes mellitus (T2DM). However, PA levels among this population remain suboptimal. This study aimed to explore patients’ perspectives on the barriers and facilitators to PA participation among Ghanaian adults with T2DM. Thirteen adults with T2DM were recruited from Korle-Bu Teaching Hospital, Ghana, for this qualitative descriptive study. Semi-structured interviews were conducted, and the data were analyzed using thematic analysis. Two overarching themes (personal factors and socio-structural factors) and 10 sub-themes relating to PA barriers and facilitators were identified. Participants had limited awareness of the recommended PA guidelines for T2DM management. Chronic illness-related factors hindered exercise participation. Difficulty differentiating between PA and exercise impeded the achievement of PA targets. Socio-structural barriers include concerns about social ridicule or embarrassment, safety during outdoor activities, a lack of culturally appropriate exercise facilities, and high social and work demands. Despite these barriers, participants were motivated by their understanding of the health benefits of PA. They emphasized integrating PA into daily routines through walking, work-related tasks, and household chores. Motivation and PA education from healthcare professionals are valued supports in achieving PA targets. Our findings showed that PA behaviour in Ghanaian adults with T2DM is influenced by both personal and external factors. Tailored PA interventions for this population should address identified barriers while leveraging facilitators to implement successful PA programs.

## 1. Introduction

Type 2 diabetes mellitus (T2DM) is a global epidemic, particularly affecting low- and middle-income countries [1]. Despite having a relatively low regional prevalence rate, sub-Saharan Africa is projected to face the highest diabetes burden in terms of both prevalence and economic impact [2]. According to the latest projections by the International Diabetes Federation (IDF), the global number of adults with T2DM is expected to rise by 51% within the next 20 years [2]. The sub-Saharan African region is projected to experience the largest increase in this diabetes prevalence, with a 129% rise from 19.4 million in 2019 to 55 million in 2045 [2]. In Ghana, a country in sub-Saharan Africa, the disease prevalence has significantly increased from approximately 0.2% in the late 1950s [3] to a current estimate of 6.5% [4], surpassing the continental average of 4.5% [2].

A self-care regimen is a priority approach in diabetes management due to its positive impact on diabetes outcomes [5]. The findings of a systematic review on self-care behavior in Ghanaians with T2DM show that adherence to a self-care regimen is suboptimal [6]. Physical activity (PA) is an essential component of a self-care regimen [7]. It helps individuals with T2DM control their blood sugar levels, lower their risk of developing diabetes-related complications, and improve their quality of life [8,9,10]. According to the American Diabetes Association and the American College of Sports Medicine, individuals with T2DM should undertake at least 150 min per week of moderate-to-vigorous aerobic exercise spread out over at least 3 days during the week, with no more than two consecutive days between bouts of aerobic activity [11]. In addition to aerobic training, at least 2–3 days/week of moderate to vigorous resistance training is recommended [11].

A systematic review, which included 34 articles, reported that PA was the self-care activity that was performed less frequently in people with T2DM [12]. In Ghana, a randomized controlled trial found that a home-based PA program was feasible and acceptable for Ghanaian adults with T2DM, and participants had high compliance and participation rates with the program [13]. According to some evidence in Ghana, exercise was the most adhered to self-care activity among Ghanaians with T2DM [14]. However, the report shows that barely 35% of Ghanaians participated in structured exercise [14]. Therefore, approaches to optimizing PA participation in this priority population are needed. There are general interventions known to effectively promote PA in people with T2DM, particularly at the community level. For example, telehealth interventions, including virtual coaching sessions and remote monitoring, offer convenient and accessible support for people with T2DM to engage in regular PA [15]. Additionally, engaging people with T2DM in peer support groups can be highly beneficial [16].

The exploration of unique population characteristics becomes imperative as we acknowledge the need to uncover the intricate nuances of each population. Recognizing the inadequacy of a “one-size-fits-all” approach [17], it becomes essential to assess specific characteristics of the population that may impact PA participation. By doing so, we can align existing scientific knowledge with the unique requirements of each population, paving the way for tailored interventions and improved outcomes.

A qualitative descriptive study conducted in Ghana examined healthcare professionals’ (HCPs) perspectives on PA behavior among Ghanaian adults with T2DM [18]. The study results show that patients often dismissed physical therapists’ advice regarding PA as they heavily relied on recommendations from doctors who rarely give PA prescriptions and guidance [18]. This negatively impacted their PA needs. Further, some patients perceived PA as irrelevant for diabetes treatment [18]. However, the study also identified several facilitators to PA participation, including addressing knowledge gaps among patients and integrating clinical exercise physiologists into usual diabetes care in Ghana [18]. Notably, that study predominantly focused on the viewpoint of HCPs, with limited attention given to understanding the experiences and perspectives of patients themselves. Consequently, there exists a critical research gap necessitating an examination of factors that impact PA behaviour specifically from the patients’ standpoint within this population. Thus, the primary objective of our study was to explore the opinions and perspectives of Ghanaian adults diagnosed with T2DM in relation to the barriers and facilitators they encounter in participating in PA.

## 2. Materials and Methods

### 2.1. Design, Setting, and Participants

Using a qualitative descriptive design, we explored the barriers and facilitators to PA participation among Ghanaian adults with T2DM. Ethics approvals were obtained from the Deakin University Human Ethics Advisory Group (Reference: HEAG-H 237_2020) and the Korle-Bu Teaching Hospital (KBTH) Institutional Review Board (Reference: KBTH-IRB/000146/2020). The principles of the Declaration of Helsinki [19] were adhered to during this study. Participants provided verbal consent prior to conducting the interview. The Standards for Reporting Qualitative Research guided this report.

Participants were recruited from the National Diabetes Management and Research Centre (NDMRC) of the Korle-Bu Teaching Hospital, the largest tertiary hospital in Ghana. To participate in this study, individuals were required to meet the following eligibility criteria: (a) being at least 18 years of age or older; (b) having a diagnosis of T2DM; and (c) residing in Ghana for at least two years. The last criterion was to ensure that the participant has a strong adherence to the Ghanaian cultural setting.

### 2.2. Recruitment and Data Collection

A research assistant displayed the study flyer at the NDMRC. A nurse or consulting physician at the NDMRC identified potential participants. Where an individual expressed interest in the study, the nurse or physician gained verbal consent for their contact details to be sent to the researchers. A research team member contacted potential participants and explained the study via telephone. Individuals who met the study criteria and were interested in taking part in this study were provided with study information. Once verbal consent was received, a convenient time for the interview was organized.

Data were collected via individual, semi-structured interviews between February and June 2021. Semi-structured individual interviews were adopted to allow participants to freely describe their experiences and perceptions about the phenomenon. The aim of the interviews was to explore factors relevant to Ghanaian adults with T2DM that influence their PA participation—a finding needed to inform the development of a future PA program. A semi-structured interview guide (Appendix A) was developed by the authors based on a review of the literature and in line with the study objectives. The interview guide was piloted with two participants. No changes were made.

Interviews were conducted by the first author (MA). The interview guide involved two sections: one focused on participants’ personal information, and the other contained questions about barriers and facilitators to PA participation in Ghanaian adults with T2DM. As part of their personal information, participants provided a subjective assessment of their PA level by categorizing themselves as either inactive, moderately active, or highly active. Moderate activities were defined as those that require a moderate level of physical exertion (e.g., biking at a regular pace), causing slightly harder breathing than usual. High-intensity activities were defined as those that demand significant physical effort (e.g., fast bicycling and heavy lifting), leading to heavier breathing than normal. Interviews regarding barriers and facilitators to PA involved open-ended questions such as ‘Tell me more about the type of exercise you do on a weekly basis’, ‘What encourages you to exercise?’, ‘Tell me more about what discourages you from exercising’, ‘Tell me about your exercise goals’, and ‘What can be done to support you to overcome exercise barriers?’ Probing questions were used to seek clarification on participants’ responses.

Interviews lasted between 30 and 60 min, and were audio recorded via Zoom. The audio recordings were transcribed verbatim by one researcher after each interview. The analysis of these interview transcripts was done alongside data collection. This approach allowed the continuous adjustment of codes as new insights emerged.

The sample size of this study was determined based on achieving theoretical data saturation. After the 13th interview, data saturation was observed. This signified a point where no new findings were emerging, implying that gathering additional data would not contribute new perspectives. Consequently, a decision was made not to continue recruitment.

### 2.3. Data Analysis

The data analysis procedure in this study followed the six phases of reflexive thematic analysis as described by Braun and Clarke [20]. This systematic approach to data analysis allows for a comprehensive understanding of the data and yields meaningful insights relevant to the research objectives [20]. Firstly, the transcripts were carefully read and analyzed to become familiar with the data. Secondly, the data were coded using NVivo 12 software [21] to identify key themes and concepts. These codes were then grouped into sub-themes based on their similarities. Thirdly, the sub-themes were reviewed by the authors to ensure accuracy and consistency. In this review, the main themes were identified and defined. Finally, the authors reached a consensus on the definitions and labels for each theme.

### 2.4. Rigour

This study adhered to the criteria proposed by Lincoln and Guba [22] to ensure methodological integrity, including credibility, dependability, and confirmability. To establish credibility, participants were selected based on predefined inclusion criteria. This ensured that participants possessed the relevant insights related to barriers and facilitators to PA participation in Ghanaian adults with T2DM. Additionally, the inclusion of verbatim quotes from participants further supported the credibility of the findings. Dependability was demonstrated by maintaining an audit trail, which served as documentary evidence of the sequential activities conducted throughout the research process. This documentation ensured that the study could be replicated and verified, enhancing the reliability of the findings. Confirmability was strengthened through independent data analysis by two of the investigators (MA and YY). Each investigator analyzed the data individually, and their interpretations were subsequently compared and discussed by all authors. This rigorous process ensured that the thematic framework and interpretations were derived directly from the data, establishing confirmability.

## 3. Results

In total, thirteen individuals with T2DM participated in this study. Participant characteristics are shown in Table 1. Ten (76.9%) females participated, and the average diabetes duration was 7.3 (SD = 8.6) years. Most of the participants (*n* = 10) lived in urban areas; three resided in rural areas. Participants were aged between 40 and 70 years. Twelve participants had at least secondary education, and only one participant had not received any formal education. Eleven (84.6%) participants described themselves as moderately active, and two (15.4%) indicated that they were physically inactive.

As shown in Table 2, there are two main themes (personal factors and socio-structural factors) and ten sub-themes relating to barriers and facilitators to PA participation were identified.

### 3.1. Personal Factors

Personal factors pertaining to Ghanaian adults with T2DM either promote or hinder their PA participation. The findings from this study show several personal factors, including a lack of understanding regarding PA guidelines for T2DM management, chronic-illness-related factors inhibiting PA participation, and difficulty distinguishing between PA and exercise. Conversely, there were personal factors that facilitated PA in the participants. These include the perceived health benefits associated with PA and the belief among patients that they can achieve their PA goals by incorporating PA into their daily routines.

Most participants demonstrated a lack of understanding regarding the guidelines and recommendations for PA in the context of T2DM management, i.e., 150 min of moderate exercise, 75 min of vigorous exercise, or a combination of both, performed each week with no two consecutive days without exercise [23]. This lack of awareness hinders their ability to engage in appropriate PA behaviors that could effectively improve their condition.


*All I do is … exercise as much as I can … But I know I’m expected to exercise for at least an hour a day [throughout the week].—CON002, female, 66 years.*


None of the participants were able to differentiate between PA and exercise. Participants’ explanations of the two terms demonstrated a weak understanding of the terms. This confusion hinders their ability to incorporate PA into their daily routines effectively and select appropriate activities to meet their needs.


*I know anything that helps me to sweat is exercise … let’s say, I weed my garden, that’s exercise. I go out for walking, that’s exercise. And as far as both [exercises] involve physical movement, I think they’re both physical activities.—CON013, female, 53 years*


Participants highlighted the influence of chronic illness-related factors on their PA participation. These factors, such as fatigue, joint or muscle pain, dizziness, and palpitations, act as obstacles to participation in PA, thereby restricting their overall levels of PA.


*I can’t exercise like the strong guys [healthy people]. At my age, my hypertension doesn’t allow me to exercise. My heart beats too fast … my knee [arthritis] doesn’t allow me [hurts when I exercise].—CON012, female, 69 years*


Most participants described their perceptions about the benefits of PA for their condition. Achieving glycemic control was the most reported benefit of PA. Other perceived benefits included improved body weight, physical fitness, and general health.


*Exercise will control my [blood] sugar. Although I don’t exercise as much as I wish, I know exercising will control my diabetes.—CON001, female, 54 years.*



*Exercise makes me light [lose weight]. I feel I’ve [got] more energy any time I get serious [with exercise] …. and in the night, you enjoy your sleep.—CON005, female, 46 years*


Six out of the 13 participants cited fear of diabetes complications, including severe neuropathy, blindness, and early death, as their sources of motivation to adopt and maintain an active lifestyle. A participant described his neuropathy in the following words:


*Last time, I was walking, and my slippers removed [from my leg whiles I was walking] and I didn’t notice … I kept walking until someone prompted me. All these happened to me because of laziness. I don’t want to die early and leave my children. I need to take my exercise serious.—CON007, male, 60 years.*


Most participants believed that incorporating PA into their daily routine would make it easier for them to achieve PA goals. Participants cited domestic chores and work-related activities that could be integrated into their daily routines to promote PA. Except for two participants, all participants identified walking as a readily accessible form of PA.


*Walking is good for me. You see it in your body when you walk [you can feel the benefit in the body]. …. I’m a Jonny Walker [enjoys walking]. I take the small boys [grandchildren] to school everyday [by walking] …… because it’s difficult to get ‘trotro’ [local name for public transport] in the morning, I walk to Kaneshi [about 2.5 km from home] every day to [get the] … ‘trotro’ to work.—CON010, male, 56 years.*


Participants also described how they could meet their PA target if they took advantage of domestic chores and work-related activities.


*With [the nature of] my work, I do a lot of up and down [walking around]. That is exercise. I also carry things a lot [at the workplace] … Even at home, I go to fetch water [from the community pipe-borne water], I wash the bowls [do dishes].—CON011, female, 47 years.*



*I can do a lot of activities at home. There’s a well in our house … every morning, I can assist to fetch the water. It’s also easy for me to pound fufu [local dish requiring moderate-to-vigorous movements to prepare]—CON004, male, 51 years.*


### 3.2. Socio-Structural Factors

Socio-structural factors related to the environment (e.g., gymnasium, community parks, walkways, healthcare system, economic system) or social relations (e.g., friends, families, HCPs) pose barriers or facilitators to PA uptake and maintenance. These factors included concerns about social ridicule or embarrassment, safety during outdoor activities, a lack of culturally appropriate exercise facilities, and high social and work demands. Additionally, motivation and PA education from HCPs were seen as valuable support for people with T2DM.

Participants expressed apprehension regarding the potential social ridicule or embarrassment associated with exercising in public. This fear acted as a barrier, discouraging their participation in outdoor exercise.


*For me, I feel shy to exercise in public places … as if the whole world is watching me….my body, people will be standing and staring at me … I would rather exercise at home.—CON003, female, 41 years*


Three out of the 13 participants expressed safety concerns while exercising outside. Participants highlighted worries about personal safety, particularly in poorly lit areas. They voiced concerns about potential robbery attacks and instances of ritualistic killings. These concerns further hinder their willingness to engage in outdoor PA during dark hours.


*It is not safe to exercise outside [especially] at dawn. The recent [serial] killings of women makes me fear going out early in the morning to exercise.—CON006, female, 46 years.*


The lack of culturally appropriate exercise facilities was another important factor identified by most participants. They expressed difficulty finding exercise facilities that catered to their cultural preferences or provided a welcoming and inclusive environment, which limited their access to suitable PA options.


*You don’t get a place to exercise here [in the community] …. even if you find a park, the young people [youth] have taken over, you don’t belong there. We don’t have more training centres [gyms], the few ones are for the macho men [muscle builders]—CON007, male, 60 years.*


Another significant barrier to PA cited by most participants was busy social and work demands. Competing responsibilities, such as family obligations or demanding work schedules, left little time and energy for engaging in PA.


*In the morning, you leave early [to work] and come back late…. You know, the pressure is too much. Weekend is good [for exercise] but there are so many [social] activities… funerals, weddings … so you keep postponing [your exercise]. …. Before you realise you’ve not done any [exercise] for the week.—CON009, female, 57 years.*


All participants commended HCPs at the NDRC for providing PA advice and motivation to encourage them to exercise.


*Any time we go for check-up [routine medical visit to the diabetes clinic], the nurses talk to us about exercise [group education] …… they encourage us. It helps us very much.—CON013, female, 53 years.*


## 4. Discussion

This study aimed to explore factors influencing PA participation from the perspective of Ghanaian adults with T2DM. Participants had limited awareness of the recommended PA guidelines for T2DM management. Chronic illness-related factors hinder exercise participation. Difficulty differentiating between PA and exercise impedes the achievement of PA targets. Socio-structural barriers included concerns about social ridicule or embarrassment, safety during outdoor activities, a lack of culturally appropriate exercise facilities, and busy social and work demands. Despite these barriers, participants were motivated by their belief in the health benefits of PA. They emphasized the integration of PA into daily routines through walking, work-related tasks, and household chores. Motivation and PA education from HCPs were valued as supports for achieving PA targets.

This study found that Ghanaian adults with T2DM recognize the positive health benefits associated with PA, consistent with findings from another study [24]. However, a significant challenge arises from their lack of comprehensive understanding regarding the recommended PA regimen, hindering their ability to achieve favorable health outcomes. Similar patterns have emerged in previous research conducted with the general adult population in Ghana [25], Ethiopian adults with T2DM [26], and Sri Lankan adults with diabetes [27]. These studies also revealed a lack of understanding among participants regarding self-care practices and meeting PA targets. While the Sri Lankan study indicated that participants were aware of the benefits of PA, they lacked understanding about the specific types, timing, intensity, and frequency of PA [27]. Furthermore, a quantitative study in Hong Kong among adults with T2DM demonstrated that a significant proportion of participants were unaware that resistance exercises, including weight lifting, were important components of recommended exercise guidelines [28]. The lack of clarity on PA guidelines, including the difference between PA and exercises, should be addressed. There is a need for greater PA awareness and education among this population. Healthcare professionals should emphasize the specific PA recommendations while considering individual needs and preferences as well as specific cultural adaptations.

The findings of this study shed light on the prioritization of family and work responsibilities over exercise among both Ghanaian men and women. This emphasis on familial and occupational duties may be influenced by cultural norms and economic circumstances. These findings align with previous studies that have identified busy household schedules as barriers to the adoption and maintenance of PA among adults with T2DM [29,30,31]. In many Ghanaian cultures, women are traditionally viewed as the primary caretakers of the household [32]. As a result, they bear significant domestic responsibilities in addition to their work outside the home, potentially limiting the time available for exercise. Similar competing priorities have been reported in studies involving individuals with T2DM [27,33]. A scoping review of barriers and facilitators for PA in adults with T2DM found that lack of time was a perceived barrier, but also noted that participants often prioritized other activities over exercise [34]. Some evidence suggests that lack of time to exercise is arguably a perceived barrier rather than an actual barrier to PA participation [35]. These findings highlight the challenges faced by individuals with T2DM in balancing multiple responsibilities and incorporating regular exercise into their routines. Understanding the influence of cultural and economic factors on exercise prioritization can inform interventions and support strategies that account for these complexities. Further, HCPs should explore patients’ competing priorities and discuss the value patients place on PA. By helping individuals navigate competing priorities, HCPs and researchers can promote the adoption and maintenance of PA among Ghanaian adults with T2DM.

A striking finding of this study is that most participants held the myth that domestic activities were enough to meet their PA target. There is evidence that PA includes various activities performed during both occupational and leisure time, such as walking, gardening, and housework, that people with T2DM can engage in to improve their health [36]. While there is no evidence that household duties are adequate to meet PA goals, there is evidence that household chores can be a beneficial form of PA for people with T2DM [36]. Thus, structured PA interventions should target patient education on the choice of domestic activities that could result in positive health impacts.

Walking was reported in this study as the most common type of exercise among participants. In a systematic review examining the effect of exercise on type 2 diabetes, Hamasaki and colleagues [36] reported that walking was the most common type of PA in people with T2DM. There is strong evidence that walking alone [37,38,39,40] or walking plus other types of exercise [13,41,42] improves cardiometabolic markers and physical fitness in adults with T2DM. However, there were concerns raised by participants in this study, suggesting that they did not feel safe exercising outdoors after dark hours. Other qualitative studies involving people with T2DM have found that individuals do not feel safe exercising outside during dark hours [43,44]. During patient consultations, HCPs should emphasize walking as a readily accessible and cost-effective form of PA. Further, discussion with patients on safe exercise hours in the community may encourage individuals to exercise when it is safer in the community.

As described by Bandura Bandura [45], people’s behaviors are influenced by expected physical or self-evaluative outcomes. Our study found that participants had a perceived range of PA outcomes, including improved blood glucose control, weight loss, increased energy, and improved overall health. These outcomes are inline with factors reported in previous studies regarding the perceived benefits of PA [34,46]. As diabetes HCPs explore cognitive factors in promoting PA in their clients, it is important to consider the perceived PA benefits identified in this study and how they can be modified to achieve positive results.

Another interesting finding in this study is that participants found advice and motivation from HCPs to be highly beneficial forms of social support. Enhancing social support in a priority population requires collaboration with HCPs, patients, and significant others [47]. Despite the advice individuals receive during routine consultations at the diabetes clinic, there appears to be a lack of understanding about PA guidelines, i.e., types, intensity, duration, and frequency. Specific strategies to enhance effective PA teaching should be considered. Role modeling has been described as a modifiable variable for enhancing exercise self-efficacy [48]. Potentially, learning can be improved by encouraging visualisation, followed by repetition of the modelled behavior [48]. This highlights the importance of building strong patient-provider relationships and incorporating supportive communication strategies into healthcare practice. Such actions may potentially enhance patient motivation, adherence to treatment plans, and overall health outcomes.

Interpretation of the findings of this study should be conducted with consideration of its limitations. Gender asymmetry is a limitation, as attempts to enroll an equal number of males and females were difficult because of the high female attendance that characterizes diabetes clinics in Ghana. Moreover, only patients who seek biomedical healthcare were recruited in this study. This recruitment process may have excluded patients who do not utilize biomedical healthcare. Such patients usually rely on alternative sources of treatment, such as spiritual healing. Mistrust of biomedical forms of treatment coupled with strong beliefs in spiritual or herbal treatment prevents some patients from accessing hospital healthcare [6,49]. Therefore, the findings should be interpreted in the context of the variety of patients with T2DM. Lastly, the assessment of the PA level of participants was based on their subjective views. Objective measurement of PA levels was beyond the scope of this study.

## 5. Conclusions

This study sheds light on the barriers and facilitators of PA participation among Ghanaian adults with T2DM, providing valuable insights from the perspective of patients. Our study found that participants with T2DM had limited awareness of recommended PA guidelines. Chronic illness-related factors hinder exercise participation, and difficulty differentiating between PA and exercise impedes PA targets. Socio-structural barriers, including social concerns, safety, a lack of suitable facilities, and busy personal schedules, were identified. However, participants were motivated by their understanding of the health benefits of PA and emphasized its integration into their daily routines. Healthcare professional support and PA education were valued. Tailored interventions should address barriers and leverage facilitators to implement successful PA programs for Ghanaian adults with T2DM.

## Figures and Tables

**Table 1 jfmk-08-00127-t001:** Characteristics of the sample.

Characteristics	N (%) [*n* = 13]
Sex	
Male	3 [23.1%]
Female	10 [76.9%]
Age, years	
40–54	7 [53.8%]
55–70	6 [46.2%]
Marital status	
Married or cohabitating	7 [53.8%]
Never married or divorced	6 [46.2%]
Employment status	
Employed	8 [61.5%]
Not employed/retired	5 [38.5%]
Education	
No formal education/primary	1 [7.7%]
At least secondary education	12 [92.3%]
Physical activity level	
Inactive	2 [15.4%]
Moderately active	11 [84.6%]
T2DM duration (years)	7.3 (SD = 8.6) *

* mean ± SD, N = total number of participants, T2DM = type 2 diabetes mellitus.

**Table 2 jfmk-08-00127-t002:** Major themes and sub-themes.

Main Themes	Sub-Themes	
	Barriers	Facilitators
Personal factors	Lack of understanding about PA guidelines for T2DM Difficulty differentiating between PA and exercise Chronic-illness-related factors inhibit engagement in exercise	Perceived health benefits motivate people with T2DM to meet their PA goals Incorporating PA into daily routine through walking, work-related activities, and household duties are important factor in achieving PA target
Socio-structural factors	Concerns about social ridicule and embarrassment Safety concerns during outdoor activity Lack of culturally appropriate exercise facilities is a barrier to exercise High social and work demands make it difficult to meet PA goals	Motivation and PA education from HCPs are valuable support for people with T2DM

PA = physical activity, HCPs = healthcare professionals, T2DM = type 2 diabetes mellitus.

## Data Availability

The data presented in this study are available on request from the corresponding author.

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
