# Peer review of "Understanding Physical Activity Behavior in Ghanaian Adults with Type 2 Diabetes: A Qualitative Descriptive Study"

_jfmk, 2023, doi:10.3390/jfmk8030127_

Round 1

Reviewer 1 Report

In this work, Amin et al. investigated the perspective of Ghanaian citizens on barriers and facilitators to participation in physical activity in adults affected by T2DM. Overall, I find the article well written and, as shown, of possible future social relevance. Certainly, the sample analyzed is very small and does not allow the generalization of what is reported to the whole nation;  i just have a few concerns and minor considerations:

- Line 77: The study found that patients often dismissed physical therapists' advice regarding PA as they heavily relied on recommendations from doctors - It would be worth specifying in the text why physical activity advice based on doctors' recommendations led to abandonment of the activity.

- Line 106:  A nurse or consulting physician at the NDMRC identified potential participants. Where an individual expressed interest in the study, the nurse or physician gained verbal consent for their contact details to be sent to the researchers. A research team member contacted potential participants, checked their eligibility for the study and explained the study via telephone - In this part of the text it is not clear who checked the inclusion criteria (whether the healthcare professional or the researcher). I suggest to remove the sentence in line 110 "checked their eligibility for the study".

- Line 129: The sample size of  the study was determined based on achieving theoretical data saturation. - This should be better explained.

- Table 1: all the abbreviations used should be reported in the notes

- Limitations section (line 366-368):  findings of this study reflect the opinion of patients who seek biomedical healthcare and does not reflect the views of a significant proportion of patients who do not seek biomedical healthcare because of economic or cultural barriers.- I really don't think it can be said that 10 patients represent the point of view of the patients who rely on biomedical healthcare. The sentence should be reformulated in a less absolutist key and the limitation of the small size of the sample should be highlighted.

As far as i concern, the text is written in good English, flowing and easy to understand. I just noticed a small error:

Line 124: ‘Tell me more about what discourages you from exercising?

Even if a request is made, i think that this is a statement and not a question; therefore, the question mark should be removed.

Author Response

Kindly see attached.

Reviewer 2 Report

The investigated file is too small and the data statistically insignificant. The interview is not clearly structured and cannot be adequately statistically objectified

Author Response

Kindly see attached.

Round 2

Reviewer 2 Report

How was the intensity of the load defined? What activities at what intensity were defined as moderate intensity physical activity? in :(Table 1. Characteristics of the sample)

Author Response

Kindly find attached
